# Afraid of the dark: Light acutely suppresses activity in the human amygdala

**Elise M. McGlashan**[ID][1☯], **Govinda R. Poudel**[2☯], **Sharna D. Jamadar**[1], **Andrew J. K. Phillips**[1], **Sean W. Cain**[1]*

**1** School of Psychological Sciences and Turner Institute for Brain and Mental Health, Monash University, Melbourne, Victoria, Australia, **2** Mary MacKillop Institute for Health Research, Australian Catholic University, Melbourne, Victoria, Australia

☯ These authors contributed equally to this work.
* sean.cain@monash.edu

**Data Availability Statement:** The first-level model outputs and scripts supporting our findings are available at https://osf.io/6ep4t/.

**Funding:** This work was funded by a Turner Institute Strategic Project Grant awarded to SWC,

## Abstract

Light improves mood. The amygdala plays a critical role in regulating emotion, including fear-related responses. In rodents the amygdala receives direct light input from the retina, and light may play a role in fear-related learning. A direct effect of light on the amygdala represents a plausible mechanism of action for light's mood-elevating effects in humans. However, the effect of light on activity in the amygdala in humans is not well understood. We examined the effect of passive dim-to-moderate white light exposure on activation of the amygdala in healthy young adults using the BOLD fMRI response (3T Siemens scanner; $n = 23$). Participants were exposed to alternating 30s blocks of light (10 lux or 100 lux) and dark (<1 lux), with each light intensity being presented separately. Light, compared with dark, suppressed activity in the amygdala. Moderate light exposure resulted in greater suppression of amygdala activity than dim light. Furthermore, functional connectivity between the amygdala and ventro-medial prefrontal cortex was enhanced during light relative to dark. These effects may contribute to light's mood-elevating effects, via a reduction in negative, fear-related affect and enhanced processing of negative emotion.

## Introduction

Beyond vision, light has powerful effects on brain function and human health. Non-visual effects of light include the regulation of the circadian clock [1, 2], alertness, physiological arousal [3], and mood [4]. These effects are likely to be largely driven by intrinsically photosensitive retinal ganglion cells (ipRGCs) containing the photopigment melanopsin, which project to many subcortical brain regions [5, 6]. In rodents, ipRGCs directly innervate the amygdala [7], a neural structure that is central to the regulation of emotion [8]. The amygdala, and connections between the ventro-medial prefrontal cortex (vmPFC) and amygdala, play a role in regulating fear-related responses [8–11]. This circuitry is critical for the regulation of negative affect.

In nocturnal animals, light is aversive. Light acutely enhances fear-related learning [12], and the retina-amygdala pathway is involved in light-related alterations in both mood and

SDJ, EMM and AJKP and a Medicine Nursing and Health Sciences Platform Access Grant awarded to SWC and GRP. GRP is supported by an ACURF Program Grant. SDJ is supported by an Australian National Health and Medical Research Council Fellowship (APP1174164). The funders had no role in study design, data collection and analysis, decision to publish, or preparation of the manuscript.

**Competing interests:** EMM, GRP & SDJ declare no relevant conflicts. AJKP and SWC are both investigators on projects funded by the Alertness Safety and Productivity CRC, have received research funds from Versalux and Delos, and, consulted for Beacon. SWC has additionally consulted for Versalux and Dyson. This does not alter our adherence to PLOS ONE policies on sharing data and materials.

learning [13]. In humans, light exposure improves mood [14] and alters function in brain areas that are important for cognition [15]. Light therapy is an effective, rapidly acting intervention for depressive disorders, including major depression and seasonal affective disorder [4, 16]. Pre-clinical work suggests that the amygdala is one of the primary brain areas which may underpin the direct effects of light on mood [13]. However, the mechanism for light therapy in humans remains unknown. Previous work has investigated the impact of light with differing spectral compositions on brain function using fMRI [17–19]. This work has shown that blue light exposure, relative to green light exposure, enhances responses to emotional stimuli, and enhances connectivity between the amygdala and hypothalamus [18], demonstrating a potential effect of light on emotional processing in humans. Here, we investigated the impact of dim-to-moderate white light exposure, relative to dark, on amygdala activity and amygdala-vmPFC effective functional connectivity in healthy adults.

## Materials and methods

This study was approved by the Monash University Human Research Ethics Committee. Participants gave written informed consent and were reimbursed for their time.

### Participants

A total of 24 young healthy adults completed the study. One participant's data were excluded due to excessive movement (>3mm) during scans, leaving a final sample of 23 (11 women) aged 18–32 years ($M = 22.35$, $SD = 3.07$). Participants were free from major medical conditions, were not taking regular prescription medications, and had no personal history of psychiatric conditions. Participants were largely classed as intermediate types on the Morningness-Eveningness Questionnaire [20] (65% intermediate, M = 54.4, SD = 6.65, no extreme morning or evening types). Mean self-reported bedtime and waketime were 23:25 h, and 7:56 h (SD = 00:52 h and 1:13 h respectively). Participants had minimal depressive symptoms measured by the Beck Depression Inventory II [21] ($M = 2.22$, $SD = 3.38$) and minimal levels of daytime sleepiness measured by the Epworth Sleepiness Scale [22] ($M = 3.7$, $SD = 2.74$). Data were collected between July and October 2018 in Melbourne, Australia (Winter–Spring).

### fMRI scan protocol

Participants were imaged using a 3T Scanner (Siemens Magnetom Skyra) with 20-channel head coils. High-resolution anatomical images of the whole brain were acquired using T1-weighted anatomical scans (TE = 2.07 ms; TR = 2.3 s; field of view: 256×256 mm; slice thickness: 1 mm). Functional images were acquired using echo-planar-imaging (TR = 2.66 s; TE = 30 ms; field of view: 220x220 mm; slice thickness: 2.5 mm; number of slices: 41; flip angle = 80, number of volumes = 180). The first five images of each session were discarded to allow for T1 equilibration.

Participants were scheduled for a functional magnetic resonance imaging (fMRI) scan beginning between 2 and 6 hours after their habitual waketime, M = 3:54 h, SD = 00:45 h. Scan timing was based on habitual sleep timing, and participants were asked to maintain their typical sleep timing on the night prior to the scan. Sleep was not objectively monitored prior to assessments. Participants arrived at the Monash Biomedical Imaging center ~1 hour before their scheduled scan time, during which they sat in a quiet waiting room and completed questionnaires (in regular room lighting).

Participants each underwent an ~30-min MRI scan to examine brain responses to dim (10 lux) and moderate (100 lux) light intensities, relative to periods of dark (<1 lux). Participants were asked to lay supine in the MRI scanner while a fiber-optic-based light delivery system

was fitted on the MRI head coil. Lights in and near the scanner room were switched off during data acquisition, as were those on the scanner (ambient room lighting <1 lux). Foam supports were used for participant comfort and to minimize movement during scans. Participants were exposed to a passive light stimulus paradigm and were asked to keep their eyes open other than normal blinking. The scan consisted of two 8-min exposure blocks with alternating 30-s periods of light and dark, and a 5-min period of darkness separating the two exposure blocks. The 10-lux exposures were always delivered first, and study staff spoke with participants during the 5-min break (in darkness) to avoid participants falling asleep during this period. Due to the binocular nature of the exposure, eye-tracking could not be used to monitor gaze during exposures.

## Light stimuli

Light stimuli were delivered in the scanner using a custom-built fiber-optic-based device [described in 23]. This consisted of a halogen light source (DC950H, Dolan-Jenner Industries, MA, USA) and metal-free fiber-optic cables (100-strand cable with 0.75-mm fibers, Optic Fibre Lighting, Sydney, AU) which transmitted light to two circular plastic diffusers (40-mm diameter) positioned ~50 mm above each eye. Light stimuli had a CCT of ~2800K ($\lambda_p$ = 655 nm) and were delivered at two intensities: ~10 photopic lux (4.3 $\mu$W/cm$^2$) and ~100 photopic lux (42.73 $\mu$W/cm$^2$) at the eye (intensity assessed using Tektronix J17 Luma Colour, Oregon, USA; spectral characteristics assessed using a MK350N Spectrometer, UPRTek, Taiwan). Daytime equivalent melanopic illuminance was ~34.67 and ~3.5 lux for the 100 lux and 10 lux conditions, respectively [24]. Tabulated spectral data are available in the (S1 Table).

## Data analysis

Images were converted from DICOM to NIFTI format using the dcm2nii tool. The fMRI data were processed and analyzed using Statistical Parametric Mapping (SPM12) in MATLAB 2016a (MathWorks, Natick MA, USA) and the FMRIB Software Library (FSL 6.0).

**MRI data pre-processing.** Pre-processing performed in SPM12 included motion-correction, slice-time correction, and normalization to a standard template. For normalization, the participant's T1 image was segmented into 3 tissue classes (grey, white, and CSF) using non-linear transformation implemented in SPM12. The resulting inverse deformation field was applied to the T2* EPIs. Standardized Montreal Neurological Institute (MNI) ICBM152 was used for normalization. Using FSL FEAT, data were spatially smoothed using a 6-mm full-width-at-half-maximum (FWHM) kernel and high-pass filtered with a 75-s filter.

**Statistical analyses.** Normalized fMRI data were analyzed using a general linear model (GLM) in FSL. The first-level GLM included (i) the BOLD activity during light compared with dark (30 s on *vs*. 30 s off) using a block-design regressor; and (ii) six motion parameters. Block design regressors were convolved with double gamma haemodynamic response functions. Separate first-level GLMs were run for each light intensity (10 and 100 lux). Two contrasts were defined: fMRI activity during light < dark (deactivation) and fMRI activity during light > dark (activation). To estimate the main effect of light, we averaged the contrast parameter estimates associated with 10 and 100 lux. Group-level significance of the parameter estimates was determined using a nonparametric one-sample permutation-based test using the randomise function in FSL (5000 Permutations). Amygdala voxels showing deactivation at $p$<0.05 were considered to be significantly suppressed by light (voxel-wise corrected for a bilateral amygdala mask obtained from WFU PickAtlas). Areas showing significant activation during light compared with dark (p<0.05, voxel-wise corrected for the whole brain) are shown in the (S1 Fig). To assess whether the 100-lux condition resulted in greater group-level

deactivation than the 10-lux condition, we used the randomise function in FSL to conduct a nonparametric paired permutation-based test (5000 permutations). Amygdala voxels showing greater deactivation during 100 lux compared with 10 lux at $p<0.05$ were considered to be significantly different (one-tailed; cluster corrected using threshold free cluster enhancement [TFCE] for a bilateral amygdala mask).

To investigate whether there was a differential interaction between the amygdala and ventro-medial prefrontal cortex (vmPFC) during light *vs*. dark, we analyzed functional connectivity using a psycho-physiological interaction (PPI) model in FSL. For the PPI model, the average BOLD fMRI time series in the amygdala within the voxels showing a significant main effect of light was obtained for each participant. Using a subject-specific PPI model, we modeled (i) light *vs*. dark activity using a block design; (ii) the activity in the amygdala; and (iii) an interaction term between the block task-related regressor and the activity in the amygdala. The PPI model was first estimated at the subject level using separate GLMs for each light level (10 and 100 lux). The contrast parameter estimate corresponding to the interaction term defined the difference in functional connectivity between light and dark conditions. To establish the interaction effect of light *vs*. dark, we averaged the contrast parameter estimates associated with 10 and 100 lux. To test whether the contrast parameter estimate of the interaction term was significantly different from zero, we used a nonparametric one-sample permutation-based test using the randomise function in FSL (5000 permutation). Any voxels within a vmPFC mask showing an interaction effect at $p<0.05$ (one-tailed; cluster corrected using threshold free cluster enhancement (TFCE) for the vmPFC mask) were considered significant. The vmPFC mask was defined using the neurosynth tool (https://neurosynth.org/), which identified vmPFC voxels functionally connected to the amygdala (seed MNI: 19, -4, 14) using a resting-state fMRI sample of 1,000 participants [25–27]. The vmPFC mask was identified by thresholding the functional connectivity map identified by neurosynth at $r>0.15$.

## Results

We found a significant ($p<0.05$, small volume voxels corrected) reduction in amygdala activity during light (averaged across conditions) compared with dark (MNI coordinate of the local maxima: 19, -4, -14; $t(22) = 8.35$; $p = 0.0002$; total number of significant voxels: 142, see Fig 1A and 1B). Light also activated the visual cortex and lateral geniculate areas of the thalamus (Fig 1). When considered separately, the 100-lux condition resulted in a significant reduction in amygdala activity (MNI coordinate of the local maxima: 21, -4, -12; $t(22) = 5.65$; $p = 0.0002$ corrected; total number of significant voxels: 364), while the 10-lux condition resulted in a below-threshold reduction in amygdala activity (MNI coordinate of the local maxima: 16, -2, -16, $t(22) = 3.5$; $p = 0.065$ corrected). There was significantly greater suppression of BOLD activity ($p<0.05$, small volume cluster corrected) due to 100 lux compared with 10 lux in the right amygdala (MNI coordinate: 25, -7, -19, $t(21) = 3.47$, $p = 0.045$, cluster size = 4 voxels, Fig 1C). We additionally found a significant ($p<0.05$, small volume voxel corrected) psychophysiological interaction effect (light > dark) of amygdala activity in the vmPFC area (local maxima MNI coordinate: 2, 35, -14; $t(22) = 3.98$; $p = 0.005$; cluster size = 53 voxels). During light there was significantly greater functional connectivity between amygdala and vmPFC activity, compared with dark. Voxels with a significant interaction are shown in Fig 1D.

## Discussion

We studied the acute effect of light on amygdala activity in humans using fMRI. Our results show that light acutely suppresses activity in the amygdala and enhances connectivity between the amygdala and vmPFC. Moderate light (100 lux) resulted in greater suppression of

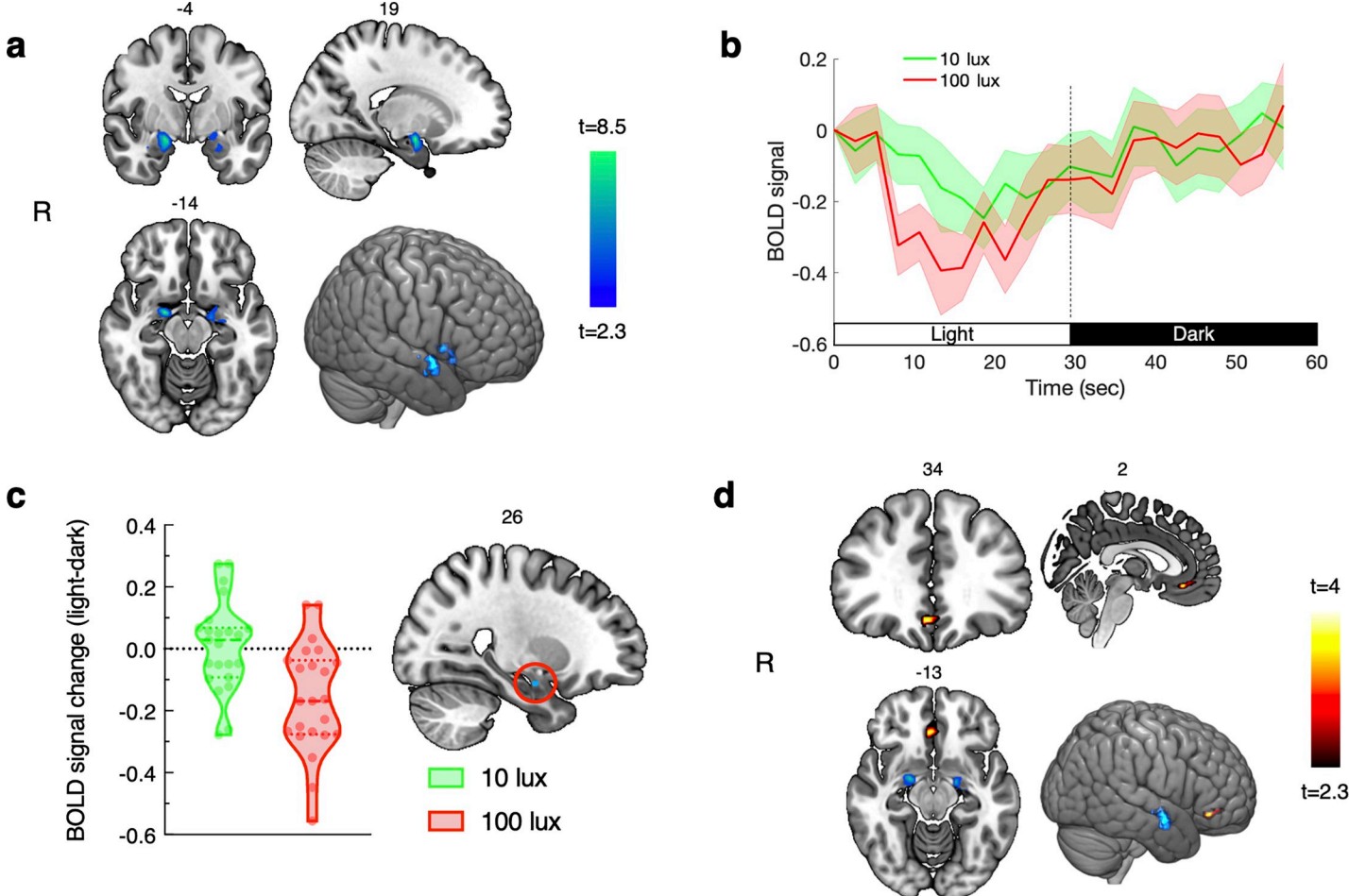

**Fig 1. Light, compared with dark, decreased activation in the amygdala and increased functional connectivity between the amygdala and ventro-medial prefrontal cortex.** (a) Voxels with significantly decreased activity in the amygdala during light relative to dark ($p<0.05$, small volume correction using bilateral amygdala mask); (b) average time-course of the baseline-corrected BOLD signal across individuals (shaded areas represent SEM). Time-courses were obtained by averaging the normalized BOLD signal for the 8 cycles each of light and dark periods from significant voxels within the amygdala; (c) BOLD signal % change in the cluster (4 voxels) showing greater deactivation during 100 lux compared with 10 lux ($p<0.05$, cluster corrected within the bilateral amygdala mask). The peak voxel was located at MNI: 25, -7, 19. Individual responses are represented by circles, and dashed and dotted lines represent the median and upper/lower quartiles, respectively; and (d) voxels (53 voxels) within the vmPFC mask showing a significant interaction effect for light *vs.* dark for functional connectivity with the amygdala ($p<0.05$, cluster corrected within the vmPFC mask). The peak voxel was located at MNI: 2, 35, -14. *Note*: Slice labels in panels a, c, and d denote MNI slice numbers. Activation maps are shown on an MNI template and visualized using MRICroGL software using radiological orientation.

amygdala activity than dim light (10 lux). These findings demonstrate a potential mechanism for improved mood with exposure to light in humans.

The amygdala and vmPFC together play a key role in the expression and regulation of fear. The amygdala is involved in the acquisition and expression of fear-related conditioning, while the vmPFC is required for effective fear extinction [28, 29]. Amygdala-vmPFC connectivity is involved in the regulation of negative affect [9], and dysfunction in this circuitry is associated with higher anxiety [30]. In nocturnal rodents, light enhances learned fear responses, both when present during acquisition and during the expression of fear responses that were learned in darkness [12]. In humans, who are diurnal, we found that light, relative to dark, is associated with an enhanced connective relationship between the vmPFC and amygdala. This suggests that light exposure may facilitate fear extinction in humans via enhanced vmPFC-amygdala connectivity. As we tested the passive response to light, accompanying behavioral and

cognitive data will provide further insight regarding the role of light in fear-related learning in humans.

There are reciprocal connections between the amygdala and vmPFC [29], and the amygdala receives direct retinal innervation in rodents [7]. Therefore, suppression of activity in the amygdala by light may permit increased prefrontal control via enhanced amygdala-vmPFC connectivity, resulting in improved regulation of affect. The potential for ipRGCs to be inhibitory is consistent with recent evidence that subtypes of ipRGCs release GABA [31]. Light interventions can be more efficacious in the treatment of depression than standard antidepressant medications [32]. Despite this, the neural mechanism for light therapy efficacy remains largely unknown. Our finding that light suppresses activation in the amygdala and enhances vmPFC connectivity, in combination with other subcortical ipRGC targets, may underpin the mechanism of action for light therapy. Assessments of brain function before and after therapeutic light interventions are needed to determine any lasting effects of increased regular exposure to bright light on mood-related brain areas.

It was previously thought that very bright light was required to elicit many of the non-visual effects of light. This was in part driven by our understanding that melanopsin has a relatively high threshold for activation [33], both in terms of intensity and duration of exposure. Previous human imaging work has shown preferential activation of cognitive brain areas in response to 'blue' light, suggesting a role of melanopsin-containing ipRGCs in mediating these responses [17, 18]. However, ipRGCs receive additional input from rods and cones (visual photoreceptors), which are sensitive to even very low levels of light [34]. Behavioral and physiological data in humans indicate that non-visual light responses can occur very rapidly, and with very dim light [35, 36]. These effects are likely due to a combination of rod, cone, and melanopsin activation. Furthermore, there are several subtypes of human ipRGCs, some of which exhibit rapid and short-lived responses [37]. Our findings and other human imaging findings [15] are consistent with rapid onset changes in brain function, which may have important implications for subsequent behavior. As we studied effects of relatively short duration stimuli, it is possible that the appearance of light (i.e., the change in visual experience) also contributed to our observed effects. It is not possible from our data to distinguish between the visual and non-visual components that may have contributed to our findings. This could be achieved by manipulating the spectral quality of the light. Further work will be needed to begin to understand the unique contribution of different subsets of ipRGCs, and other photoreceptors, to both the visual and non-visual aspects of light responses.

The ability to easily control our light environment is a very recent development in our evolutionary history. Prior to the invention of electric lighting, light exposure was largely determined by the rise and fall of the sun. The prevalence of our self-exposure to light at night in modern society may be partly motivated by the rewarding and mood-elevating effects of light. Parallel to our own findings, it has been shown that the habenula, a brain structure involved in reward regulation [38, 39], is acutely suppressed by light in humans [40]. Decreased habenula activity is associated with increased expectation of reward [38]. Our findings dovetail with preclinical evidence that the amygdala and habenula are critical to light-related mood and learning effects [41, 42]. In humans, increased mood and reward sensitivity could lead to increased light-seeking behavior at times of day when light is disruptive to the circadian system (e.g., in the evening/night). This disruption may be more severe in populations who are hypersensitive to the non-visual effects of light, including those taking medications that increase light sensitivity [43], or with certain sleep [44] or mood disorders [e.g., bipolar disorder; 45]. Conversely, low light sensitivity, which is reported in other mood disorders [e.g., unipolar or seasonal depression; 46, 47], may directly contribute to negative affect via a decreased ability of light to suppress amygdala activity.

Light is an effective therapeutic tool for mood problems. We have shown that dim-to-moderate light suppresses amygdala activation and enhances amygdala-vmPFC connectivity. These effects may contribute directly to the mood-elevating effects of light via improved emotional processing, and a reduction in fear-related emotion.

## Supporting information

**S1 Fig. Light, compared with dark, increases activity in the visual cortex and lateral geniculate.** Increased activity ($p < 0.05$, voxel-wise corrected) during light compared to dark was observed in the visual cortex (local maxima MNI coordinate: 14, -90, 2, tmax = 8.98) and right lateral geniculate area (tmax = 6.3, MNI coordinate: 26, -25, -2). Bilateral lateral geniculate area showed activation at $p < 0.001$. However, this did not survive whole-brain correction for significance. Voxels with increased activity in the visual cortex and lateral geniculate area of the thalamus are shown. Green voxels represent the voxels significant at $p < 0.05$ (voxel-wise corrected). Red-yellow color represent increased activity at $p < 0.001$, uncorrected. The brain images are presented on a radiological orientation.
(TIFF)

**S1 Table. Spectral power distribution for the light source.** Data are shown for a measure of the light source at ~100 photopic lux in 1 nm bins from 360nm to 760nm. Data are reported in $\mu W/cm^2$, measured using a MK350N Spectrometer (UPRTek, Taiwan).
(XLSX)

**S1 Data.**
(XLSX)

## Acknowledgments

We thank the Monash Instrumentation Facility for their assistance with the construction of our light delivery system. We also thank the staff and students of the Monash Biomedical Imaging center for their help with imaging data acquisition, and our participants for their time and effort.

## Author Contributions

**Conceptualization:** Elise M. McGlashan, Govinda R. Poudel, Sharna D. Jamadar, Sean W. Cain.

**Data curation:** Elise M. McGlashan.

**Formal analysis:** Elise M. McGlashan, Govinda R. Poudel, Sharna D. Jamadar, Andrew J. K. Phillips.

**Funding acquisition:** Elise M. McGlashan, Govinda R. Poudel, Sharna D. Jamadar, Andrew J. K. Phillips, Sean W. Cain.

**Investigation:** Elise M. McGlashan.

**Methodology:** Elise M. McGlashan, Govinda R. Poudel, Sean W. Cain.

**Project administration:** Elise M. McGlashan.

**Supervision:** Sean W. Cain.

**Visualization:** Govinda R. Poudel.

**Writing – original draft:** Elise M. McGlashan, Govinda R. Poudel, Sharna D. Jamadar, Andrew J. K. Phillips, Sean W. Cain.

**Writing – review & editing:** Elise M. McGlashan, Govinda R. Poudel, Sharna D. Jamadar, Andrew J. K. Phillips, Sean W. Cain.

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
