## [Decision Letter · Decision Letter 0]

1 Mar 2021

PONE-D-21-01352

Afraid of the dark: Light acutely suppresses activity the human amygdala

PLOS ONE

Dear Dr. Cain,

Thank you for submitting your manuscript to PLOS ONE. After careful consideration, we feel that it has merit but does not fully meet PLOS ONE’s publication criteria as it currently stands. Therefore, we invite you to submit a revised version of the manuscript that addresses the points raised during the review process.

Please address all suggestions made by reviewers. 

We look forward to receiving your revised manuscript.

Kind regards,

Tudor C Badea, M.D., M.A., Ph.D.

Academic Editor

PLOS ONE

"I have read the journal's policy and the authors of this manuscript have the following competing interests: EMM, GRP & SDJ declare no relevant conflicts. AJKP and SWC are both investigators on projects funded by the Alertness Safety and Productivity CRC, have received research funds from Versalux and Delos, and, consulted for Beacon. SWC has additionally consulted for Versalux and Dyson."

Reviewers' comments:

Reviewer's Responses to Questions

**Comments to the Author**

1. Is the manuscript technically sound, and do the data support the conclusions?

Reviewer #1: Yes

Reviewer #2: Yes

2. Has the statistical analysis been performed appropriately and rigorously? 

Reviewer #1: Yes

Reviewer #2: I Don't Know

3. Have the authors made all data underlying the findings in their manuscript fully available?

Reviewer #1: Yes

Reviewer #2: Yes

4. Is the manuscript presented in an intelligible fashion and written in standard English?

Reviewer #1: Yes

Reviewer #2: Yes

5. Review Comments to the Author

Reviewer #1: McGlashan and colleagues explore the relationship between light and mood by examining the effects of light on the amgydala using BOLD fMRI in healthy human participants. They find that light suppresses the activity of the amygdala (compared with dark), and enhances the functional connectivity between the amygdala and VMPFC. The study appears to be well designed, has been performed and analysed appropriately. I don’t have any major concerns with the manuscript, though there are some areas where I believe further discussion/interpretation is required. These are highlighted below:

Does 10lux actually drive a significant reduction in BOLD signal? Figure 1c suggests not (I cannot seem to find associated stats to compare light and dark for only 10lux/100lux). It is important to know whether 10lux drives a lower amplitude response; or whether 10lux is below threshold.

I appreciate there are differences in methodologies etc., but how do the authors reconcile their data and the findings of Vanderwalle (2007), who I believe show an activatory - rather than inhibitory - role for (blue) light?

The response time-course is relatively swift. Is there evidence that adjustments in emotional responses can occur in these time frames? Relatedly, do the authors observe any cumulative effects of light stimulation over the repeated presentations?

Can the authors comment on the circuitry by which light may suppress activity in the amygdala – ipRGCs are typically thought of showing excitatory responses following light activation.

The final sentence - perhaps a little too much conjecture (!).

Reviewer #2: Referee’s comments for authors:

McGlashan et al report results of an fMRI study assessing changes in BOLD activity in the amygdala over repeated exposures to 30s exposure to dim or moderate light. They report a dose-dependent reduction in activity in amygdala by light across this range. They also report an increase in connectivity between amygdala and ventro-medial prefrontal cortex. The results are interpreted in the context of evidence of light-dependent modulations in mood and the retinal projection to the amygdala reported in non-human species.

I am poorly qualified to comment on the quality of the fMRI data or its analysis and interpretation. I restrict my comments to those aspects relating to the light stimulus and in its application. The light stimulus is poorly described. The authors should provide details of the type of light source used, and also its color (in color coordinates or correlated color temperature). The authors correctly describe their stimulus in terms of melanopic illuminance, but the precise measure used and associated unit are missing (I assume they report ‘melanopic equivalent daytime illuminance’ with the unit of lux?).

Implicit in the interpretation of the outcome is that the effects observed are response to the presence of light. Given the timeframes involved it is at least a likely that they are a response to the appearance of light (suddenly switching on a light for a subject previously in the dark is a particular event in itself). This distinction may seem esoteric, but it has important consequences for the relationship between the current light stimulus and light regulation of mood. If the amygdala response is elicited by light appearance then it is not obviously related to light therapy impacts on mood. The timecourse of BOLD if anything supports the light appearance origin (although other explanations are possible). I would like to see the paper re-written with this distinction in mind. The findings here are consistent with vision impacting amygdala in humans, but content relating to light effects on mood should be toned down (including in the abstract). I recommend replacing

‘light’ with ‘light pulses’ in the title and abstract to make the nature of the stimulus clearer. Also the authors should include a discussion of the distinction between the presence vs appearance of light as an origin for their effects and the implications for interpretation.

Minor:

There is a word (‘in’ or ‘of’) missing in the title.

6. PLOS authors have the option to publish the peer review history of their article (what does this mean?). If published, this will include your full peer review and any attached files.

Reviewer #1: No

Reviewer #2: No

---

## [Author Response · Author response to Decision Letter 0]

6 Apr 2021

Reviewer #1:

Does 10lux actually drive a significant reduction in BOLD signal? Figure 1c suggests not (I cannot seem to find associated stats to compare light and dark for only 10lux/100lux). It is important to know whether 10lux drives a lower amplitude response; or whether 10lux is below threshold.

The main findings in the paper are reported for the main effect of light (averaged 10 lux and 100 lux data), we have also conducted a voxel-wise analysis of the effect of light at 10 lux and 100 lux separately. For the 100 lux condition alone, activity was significantly suppressed (p=.0002, small volume corrected for bilateral amygdala). For the 10 lux condition, the suppression was significant only at an uncorrected level (p=0.01), corrected p=.065. These findings are now included in the manuscript (line 177 onwards).

I appreciate there are differences in methodologies etc., but how do the authors reconcile their data and the findings of Vanderwalle (2007), who I believe show an activatory - rather than inhibitory - role for (blue) light?

As the reviewer states, there are methodological differences between our work and that of Vandewalle et al., (2007, PLOS ONE). In that paper, the authors compare responses to different monochromatic light sources, rather than between broad spectrum light and dark. Furthermore, the light sources they used were all quite visually dim. In terms of photopic illuminance, our 10-lux condition is the most comparable, and our result was below threshold at that light level. 

Vandewwalle et al., (2007) reported increased activation in the right amygdala only with the onset of blue light relative to green. This was not observed when the onset of blue light was compared to other monochromatic sources. This effect was transient in nature, that is, occurred only with the onset of the light stimuli and not as a sustained effect during blue light exposures. The authors suggest that this may reflect differences in response latencies between photoreceptors, where melanopsin expressing RGCs may be involved in regulating transient responses, while other photoreceptors contribute to the sustained effects. We observed a persistent rather than transient effect, at a light intensity which was visually much brighter. Given the significant differences in design (comparing different colours rather than light and dark) and light intensity, speculation on the mechanisms that may underly our results is difficult. However, this likely is primarily related to the contribution of different photoreceptors to different aspects of the response (transient vs. sustained), and it may be that effect observed by Vandewalle and colleagues is specific to blue light. 

The response time-course is relatively swift. Is there evidence that adjustments in emotional responses can occur in these time frames? Relatedly, do the authors observe any cumulative effects of light stimulation over the repeated presentations?

Unfortunately, it is not possible to assess whether there are cumulative effects of light across the repeated presentations. fMRI signals are always subject to drift over time, and so they require detrending before analysis is performed at the block level. Any cumulative effects of repeated presentations therefore cannot be distinguished from drift in the fMRI signal.

To our knowledge, there are no studies looking at changes in subjective mood with light exposures this short, but many non-visual responses such as changes in heartrate, temperature and arousal do occur very rapidly (Prayag, Jost, Avouac, Dumortier, & Gronfier, 2019). We suggest that our observed changes represent a non-conscious change which may ultimately lead to differences in affective experiences. Although limited, there are findings from fMRI studies showing that acute, short time-scale changes in amygdala activity do relate to real-world experience of emotion. For example, persistence in amygdala activity following the presentation of negative affective stimuli is associated with decreased positive affect and increased negative affect in daily life (Puccetti et al., 2021). 

Prayag, A. S., Jost, S., Avouac, P., Dumortier, D., & Gronfier, C. (2019). Dynamics of Non-visual Responses in Humans: As Fast as Lightning? Frontiers in Neuroscience, 13(126). doi:10.3389/fnins.2019.00126

Puccetti, N. A., Schaefer, S. M., van Reekum, C. M., Ong, A. D., Almeida, D. M., Ryff, C. D., . . . Heller, A. S. (2021). Linking Amygdala Persistence to Real-World Emotional Experience and Psychological Well-Being. The Journal of neuroscience, JN-RM-1637-1620. doi:10.1523/JNEUROSCI.1637-20.2021

Can the authors comment on the circuitry by which light may suppress activity in the amygdala – ipRGCs are typically thought of showing excitatory responses following light activation.

Although ipRGCs have traditionally been thought to be excitatory, recent evidence demonstrates that there is a subtype of ipRGCs which release inhibitory GABA (Sonoda et al., 2020). Furthermore, it has been shown that distinct populations of ipRGCs project to distinct brain areas to control different non-visual effects of light (Rupp et al., 2019). Therefore, it is plausible that there is a subtype of ipRGCs which project to the amygdala in humans and are inhibitory. In line with our findings, a recent paper found that light exposure suppressed activity in the human habenula, a brain area involved in mood and reward regulation (Kaiser et al., 2019). Although specific retina-brain pathways have not been as well characterised in humans as in animal models, it is highly likely that some projections are inhibitory in nature, and that this explains our finding. We have commented on this potential circuitry in the paper from line 230.

Kaiser, C., Kaufmann, C., Leutritz, T., Arnold, Y. L., Speck, O., & Ullsperger, M. (2019). The human habenula is responsive to changes in luminance and circadian rhythm. Neuroimage, 189, 581-588. 

Rupp, A. C., Ren, M., Altimus, C. M., Fernandez, D. C., Richardson, M., Turek, F., . . . Schmidt, T. M. (2019). Distinct ipRGC subpopulations mediate light’s acute and circadian effects on body temperature and sleep. eLife, 8. doi:10.7554/elife.44358

Sonoda, T., Li, J. Y., Hayes, N. W., Chan, J. C., Okabe, Y., Belin, S., . . . Schmidt, T. M. (2020). A noncanonical inhibitory circuit dampens behavioral sensitivity to light. Science, 368(6490), 527-531. doi:10.1126/science.aay3152

The final sentence - perhaps a little too much conjecture (!).

We have removed this sentence. 

Reviewer #2: 

The light stimulus is poorly described. The authors should provide details of the type of light source used, and also its color (in color coordinates or correlated color temperature). The authors correctly describe their stimulus in terms of melanopic illuminance, but the precise measure used and associated unit are missing (I assume they report ‘melanopic equivalent daytime illuminance’ with the unit of lux?).

We had described the light source in detail, but this appears to have been missed by the reviewer. The light source is a Halogen bulb, with a CCT of 2800K (included in line 109 onward). Light was delivered using a fibre optic system and controlled using custom a MatLab script. Intensity of the light source was described in terms of photopic illuminance, and irradiance (in µW/cm²), and we supplied tabulated spectral data as a supplement. 

Melanopic equivalent daytime illuminance is now reported in the text, having been determined using the updated CIE guidelines (Commission Internationale de l'Eclairage [CIE], 2018). 

Implicit in the interpretation of the outcome is that the effects observed are response to the presence of light. Given the timeframes involved it is at least a likely that they are a response to the appearance of light (suddenly switching on a light for a subject previously in the dark is a particular event in itself). This distinction may seem esoteric, but it has important consequences for the relationship between the current light stimulus and light regulation of mood. If the amygdala response is elicited by light appearance then it is not obviously related to light therapy impacts on mood. The timecourse of BOLD if anything supports the light appearance origin (although other explanations are possible). I would like to see the paper re-written with this distinction in mind. The findings here are consistent with vision impacting amygdala in humans, but content relating to light effects on mood should be toned down (including in the abstract). I recommend replacing ‘light’ with ‘light pulses’ in the title and abstract to make the nature of the stimulus clearer. Also the authors should include a discussion of the distinction between the presence vs appearance of light as an origin for their effects and the implications for interpretation.

We thank the reviewer for this point and agree that it is reasonable to suggest that our result is also related to the change in visual experience which is inherent in the presentation of light. We note that previous work has observed transient effects related to the onset of light (discussed above), as distinct from sustained effects across light exposure periods of similar durations (~30 seconds). Therefore, although our light exposure durations are relatively short, they are likely sufficient to produce responses which are due to the presence of light, rather than only the appearance of light. Nonetheless, we have edited the paper to acknowledge this alternate interpretation and agree that investigating these effects under different stimulus durations will be an important thing to consider in future research (line 249 onwards). 

Regarding the reviewers point about ‘light pulses’, we have elected not to make this change. This is because we do not feel that this would add clarity about the nature of the stimuli. Depending on the field, the term ‘light pulse’ can convey a very different meaning. For example, in circadian rhythms a 6.5-hour light exposure may be described as a light pulse, whereas if we are considering pupil responses, a light pulse may be as short as milliseconds. Therefore, we do not feel that the word “pulse” would aid the reader. We have reported the duration of the exposures in the abstract for clarity, and the timescale is outlined in the methods. 

There is a word (‘in’ or ‘of’) missing in the title.

This typo has been corrected, thank you.

---

## [Decision Letter · Decision Letter 1]

14 May 2021

Afraid of the dark: Light acutely suppresses activity in the human amygdala

PONE-D-21-01352R1

Dear Dr. Cain,

We’re pleased to inform you that your manuscript has been judged scientifically suitable for publication and will be formally accepted for publication once it meets all outstanding technical requirements.

Kind regards,

Tudor C Badea, M.D., M.A., Ph.D.

Academic Editor

PLOS ONE

Additional Editor Comments (optional):

Reviewers' comments:

Reviewer's Responses to Questions

**Comments to the Author**

1. If the authors have adequately addressed your comments raised in a previous round of review and you feel that this manuscript is now acceptable for publication, you may indicate that here to bypass the “Comments to the Author” section, enter your conflict of interest statement in the “Confidential to Editor” section, and submit your "Accept" recommendation.

Reviewer #1: All comments have been addressed

Reviewer #2: All comments have been addressed

2. Is the manuscript technically sound, and do the data support the conclusions?

Reviewer #1: Yes

Reviewer #2: Yes

3. Has the statistical analysis been performed appropriately and rigorously? 

Reviewer #1: Yes

Reviewer #2: Yes

4. Have the authors made all data underlying the findings in their manuscript fully available?

Reviewer #1: Yes

Reviewer #2: Yes

5. Is the manuscript presented in an intelligible fashion and written in standard English?

Reviewer #1: Yes

Reviewer #2: Yes

6. Review Comments to the Author

Reviewer #1: Thanks to the authors for addressing my previous comments. I have no further queries with the manuscript.

Reviewer #2: The authors have addressed my comments adequately. This work is now suitable for publication in my opinion.

7. PLOS authors have the option to publish the peer review history of their article (what does this mean?). If published, this will include your full peer review and any attached files.

Reviewer #1: No

Reviewer #2: No